# Electric Field Detection System Based on Denoising Algorithm and High-Speed Motion Platform

**DOI:** 10.3390/s22145118

**Published:** 2022-07-07

**Authors:** Qi Liu, Zhaolong Sun, Runxiang Jiang, Jiawei Zhang, Kui Zhu

**Affiliations:** 1College of Electrical Engineering, Naval University of Engineering, Wuhan 430033, China; cantonlq@163.com (Q.L.); brucesunzl@126.com (Z.S.); nue_zeek@163.com (K.Z.); 2College of Weaponry Engineering, Naval University of Engineering, Wuhan 430033, China; gaweizhang@163.com

**Keywords:** high-speed boat, low frequency electric field, noise, ICEEMDAN, denoising

## Abstract

Effective denoising can ensure fast and accurate target detection. This paper presents an electric field measurement system based on a high-speed motion platform, which was built to analyze the characteristics of low frequency electric field noise. An offshore test has shown that it is possible to detect a low-frequency electric field using a high-speed motion platform. Low frequency electric field noise was then collected to analyze its characteristics in terms of time and frequency domains. Based on the noise characteristics, complete ensemble empirical mode decomposition with adaptive noise (ICEEMDAN) was improved and combined with an adaptive threshold algorithm for denoising and reconstructing target containing noise signals. As revealed in the results, the proposed algorithm achieved highly effective denoising to overcome the line spectrum detection failure resulting from a high-speed motion platform. The detection range had also been improved from the original 853 m to 1306 m, a 53.1% increase.

## 1. Introduction

Underwater mobile platforms including autonomous underwater vehicles (AUVs), unmanned underwater vehicles (UUVs) and underwater gliders have been widely applied in marine environment monitoring and underwater target detection because of their advantages such as small size, low energy consumption, long endurance, great adaptability and good maneuverability [1,2]. In terms of underwater target detection, a passive acoustic detection system based on AUVs, UUVs and other platforms returns low power consumption and great adaptability [3,4,5]. For this reason, studies in the field of underwater target detection have currently focused on the use of mobile platform detection devices for target detection [6,7].

Currently, many underwater mobile platforms are often equipped with sonar [8]. Sonar is an acoustic device capable of emitting a signal and, through the reflected echo, of identifying an object [9]. It allows for obtaining measurements of distances between the source and the first object encountered in the direction of emission of the wave [10]. In the time domain, due to the finite speed of propagation of the acoustic perturbation, the time taken to reach increasingly distant objects is proportional to the distance traveled. It is therefore possible to go back from time information to distance information. The signal detected by the receiver in a time interval depends on the medium crossed in the delimited layer by the distances reached between two consecutive time instants, with intensity proportional to the volumetric diffusion coefficient [11]. Early studies had been conducted on the embedding of acoustic sensors into an underwater mobile platform, and some remarkable findings had been obtained in various countries, particularly in the United States. Webb proved the feasibility of equipping an AUV with sonar for gathering the acoustic signals of its target [12]. Moore performed a test with Seaglider [13], an underwater acoustic glider developed in the United States, to collect the calls of blue whales, humpback whales and sperm whales. In this US project, omnidirectional broadband hydrophones (5 Hz to 30 kHz) were mounted on the wings of the glider. Grund used acoustic sensors installed on fixed and mobile underwater platforms for coastal surveillance with the aim of detecting and tracking diesel submarines [14]. UUVs and gliders equipped with acoustic and environmental sensors are used. Lv studied the random fluctuation of the sea surface which is a difficult environmental factor to quantify [15]. It is characterized by parameters such as wind speed, wind direction and water depth in real time. In China, the Ocean University of China and the Institute of Acoustics, Chinese Academy of Sciences, jointly developed an acoustic underwater vehicle to achieve its single node detection range no less than 3 km [16]. The Navy Submarine Academy of the People’s Liberation Army and Tianjin University developed a prototype following the principles of the Dolphin underwater acoustic glider [17], which preliminarily offered autonomous target detection.

Silent submarines have been developed to significantly lower the sound source level of targets, making it more difficult to identify with single acoustic detection. However, a single passive acoustic detection method was disturbed by a high rate of false alarms and easy interference from dummy decoys. The electric fields emitted by a ship can be exploited for underwater detection. Galvanic currents flowing in the water around the hull generate an underwater electric field. This field is responsible for the extremely low frequency emission [18]. The electric field emitted by the ships had advantages such as low frequency (1–7 Hz), long transmission distance and remarkable characteristics of the line spectrum [19]. In addition to passive sensors, this method has become a reliable source of signals for tracking underwater weapons and targets. The study of low-frequency electric field sensing began very early outside of China. Birsan argued that a low frequency electric field was basically generated by the cyclic rotation of the helices [20]. Zolotarevskii built a low frequency electric field measurement platform [21]. In underwater environments, the electric field signal is more stable than the acoustic signal. Furthermore, acoustic invisibility-based methodologies make electric field sensing techniques preferred for detecting or tracking targets [22,23,24,25]. However, few studies on low frequency electric field sensing systems for underwater mobile platforms have been released to the public. In [26], regarding the detection of the low frequency electric field, the buoys were used as platforms for the detection of the electric field. A fixed electric field electrode array was used to detect and locate underwater targets in [27]. An underwater low frequency electric field sensing device was developed in [28].

In this paper, a low frequency electric field detection system based on a high-speed boat was initially devised and built. An offshore test is presented to verify the feasibility of detecting low frequency electric field of targets using a high-speed boat equipped with electric field sensors. Subsequently, noise characteristics were extracted without a target source to analyze the distribution of noise and discover the reasons for the line spectrum detection failure as shown in the test results. Next, a denoising algorithm was developed based on the distribution of noise. On this basis, an adaptive denoising algorithm is proposed for low frequency electric field target detection of high-speed motion platforms. The proposed algorithm combines improved complete ensemble empirical mode decomposition with adaptive noise (ICEEMDAN) with an adaptive threshold. The modes obtained from the ICEEMDAN will be used for threshold calculation and adaptive layer screening. In the end, measured signals will be employed to verify the performance of the proposed algorithm.

The structure of the paper is the following. Section 2 describes in detail the methodology for the identification of the low frequency electric field emitted by an underwater mobile platform. Section 3 presents a complete analysis of the noise source, through a test without a target source. The noise caused by the speed variation was collected. Section 4 analyzes the characteristics of the electric field noise emitted by the mobile platform. Section 5 presents an adaptive low frequency electric field denoising algorithm based on ICEEMDAN method with threshold. Section 6 presents the results of the methodology described in the previous sections. Finally, the conclusions and future work are reported in Section 7.

## 2. A Low Frequency Electric Field Detection System with a High-Speed Boat as the Platform

### 2.1. System Design and Construction

In this paper, a high-speed boat was used as the motion platform for electric field detection. The boat was 10.5 m long and 2 m wide, and its propulsion system used pump injection propulsion. The whole hull is made of glass steel, and only the stern water inlet and spray nozzle are made of metal. The velocity of the boat was adjustable up to 50 kn, and flexibly was set as needed during the test.

A set of orthogonal electric field sensors was provided for the bow and stern, respectively, to study the electric field noise characteristics and determine the detection range of electric field sensors at different positions of the boat as shown in Figure 1.

First, a cruciform acquisition system based on a fixed bracket is designed, as shown in Figure 2a. Each set of sensors contained four Ag/AgCl poles. This electrode, at a temperature of 25 °C and a pressure of 1 atm, has a potential equal to ΔE = +0.198 V (sensitivity = 10^−7^ (V/m); Noise floor = 3.1 × 10^−9^ V/sqrt(Hz) at 1 kHz). Each set of electrodes was mounted onto a rigid cross to measure the longitudinal electric field $E_{x}$ and transverse electric field $E_{y}$. The electrodes were spaced 0.5 m apart. Since the bottom of the actual speedboat is a non-planar structure, based on the configuration of the cross-shaped acquisition system, the fixed bracket is removed and the sensor is laid on the bottom of the speedboat in accordance with Figure 2b. Its detection principle is consistent with that of the cross-shaped acquisition system.

To ensure adherence to the moving boat, the sensors and associated cables were attached to the surface of the hull using strong glue. The cables have been protectively sheathed to prevent the electric field measurement system from turning off during the movement of the vessel, to maintain its adhesion with the movement of the vessel, and to avoid signal interference caused by any shaking irregular sensor. The sensors were mounted on the hull of the boat as shown in Figure 2b.

### 2.2. A Low Frequency Electric Field Detection Test with a High-Speed Boat Platform

A standard simulation source was taken as the target source in a test to quantitatively analyze the electric field sensing range of a moving carrier platform. Table 1 lists its parameters. The schematic diagram of the test is given in Figure 3.

Figure 3 shows that the motorboat moves in the direction of the simulated radiation source placed on the shore, the blue area is the sea area, the gray area is the shore, 13 m indicates that the electric dipole moment of the simulated source is 13 m, and the thick black arrow indicates that during the test, the speedboat moves towards the simulated source. The sampling frequency of the electric field measurement system was 250 Hz. The measurement bandwidth was set to [0.05 Hz, 60 Hz]. The water depth in the test area was around 5 m. The conductivity of seawater was 2.8 (S/m). The boat was set to sail in the longitudinal direction. A set of sensors near the bow measured the longitudinal and transverse components, *E_x_*_1_ and *E_y_*_1_, respectively, while a set of sensors near the stern measured the longitudinal and transverse components, *E_x_*_2_ and *E_y_*_2_, respectively.

During the test, the boat carrying the measurement system moved sequentially at the set speeds of 5 kn, 10 kn, 15 kn, 20 kn, and 25 kn. At each speed, the boat’s movement was kept unchanged for 380 s. With the method specified in [28], the detection of the line spectrum was carried out using the collected signals. The results of the detection of the spectrum of lines using the signals measured at the speed *v* = 10 kn are shown in Figure 4.

Figure 4a,c,e,g shows the simulated source radiation signal measured in the experiment; the abscissa is time (s) and the ordinate is *E* (uV/m) indicating the electric field intensity. *E_x_*_1_ indicates the longitudinal component of the bow sensor, *E_y_*_1_ indicates the transverse component of the bow sensor, *E_x_*_2_ indicates the longitudinal component of the stern sensor and *E_y_*_2_ indicates the transverse component of the stern sensor. Figure 4b,d,f,h shows the result of target line spectrum extraction using the method in [28]; the ordinate represents the time axis of measurement (s) and the abscissa represents the frequency of extraction (Hz). At 10 knot, the speed of the boat is not high, and the speed of the water flow is mainly along the longitudinal direction. The transverse electric field in the bow is away from the engine, which is placed in the stern, so there is no interference caused by the splashing of the water due to the high speed of the boat. Therefore, the noise is minimal, and the detection result is the best.

As shown in Figure 4, a vessel-mounted measurement system ensures effective detection of a target’s low frequency electric field signals. The target line spectrum can be successfully extracted from the electric field signals collected fore and aft, verifying the feasibility of low frequency electric field detection with a high speed boat. For comparison, the quality of the detection results was measured by a descriptor, such as the detection ratio of the line spectrum (*K*). The maximum detection time for a target’s line spectrum was $T_{\max}$, and the total motion time of a high-speed boat was $T_{total}$. With $T_{\max}$, we specify the maximum time necessary to effectively detect the line spectrum of the target. Since the spectrum of the lines is not always continuous, it can be divided into several fragments. The sum of these contributions of the detection times gives the maximum total time necessary to correctly detect the target.

The calculation is as given below:(1)$$K=\frac{T_{\max}}{T_{total}}\times 100\%=\frac{T_{\max}}{380}\times 100\%$$

The measurement results of the detection of the electric field of a target by the measuring platform moving at low to high speeds were compared with the detection results of the spectrum of lines, as shown in Table 2.

As revealed in the test results, the noise resulting from the variation in the velocity of the platform severely affected line spectrum detection, so that the detection ratio varied differently with velocity at the bow and stern. When the velocity reached 20 kn, line spectrum detection failed with the longitudinal component at the bow *E_x_*_1_.

## 3. A Noise Collection Test with a High-Speed Boat as the Platform

For the comprehensive analysis of the noise source, a test was carried out without a target source. The noise caused by the variation of velocity was collected. The same measurement system was employed as specified in Section 1.

To ensure consistency between the measurement environment and the status of the measurement system affected by the noise caused by the movement of the platform, a test has been designed as follows: the boat moved constantly at the initial velocity of 5 kn for 110 s, and then sped up to 10 kn and maintained this velocity for the same period. In the same way, it moved at the velocities of 15 kn, 20 kn, and 25 kn, each for 110 s. Noise was continuously collected at these velocities. The results are shown in Figure 5.

Based on the time domain distribution of the measured noise, it was preliminarily found that the total noise level at the bow was always lower than that at the stern. When the speed was 25 kn, the bow electric field noise peaked, but the total noise level was still very low.

## 4. Analysis of Noise Characteristics of a High-speed Boat as the Platform

### 4.1. Time Domain Characteristics of Noise Signals

To fully characterize the electric field noise in the time domain, the dimensional and non-dimensional temporal parameters of the signals were calculated. The selected dimensional characteristics included peak-to-peak value $E_{p-p}$, mean value $\sigma^{2}$ and root mean square $X_{rms}$; the selected dimensionless characteristics included kurtosis $K_{4}$ and waveform factor $S_{f}$. The calculated parameters for the characteristics of noise signals are presented in Table 3.

Based on the results given in Table 3:
The peak-to-peak value $E_{p-p}$ revealed that the noise level at the bow was lower than that at the stern when the velocity was not higher than 15 kn; nevertheless, it was higher than the latter when the velocity exceeded 15 kn.The mean value *µ* of noise at the bow and stern was always approximate to 0 at all velocities.The variance $\sigma^{2}$ showed that the noise collected at the bow and stern fluctuated very slightly at all velocities.With the root mean square $X_{rms}$, it was revealed that the effective value of noise at the bow increased with increasing velocity, but that at the stern, it increased, decreased and increased again as velocity increased, but was lowest at the velocity of 15 kn.With the kurtosis $K_{4}$, it was found that, for the same sensor, the noise at the bow peaked when the velocity was 25 kn, but the noise at the stern was distributed most gently when the velocity was 15 kn; for the same velocity, the noise at the bow was distributed more gently than that at the stern when the velocity was not greater than 20 kn.The waveform factor $S_{f}$ revealed that the noise was not white noise except the longitudinal component of the noise at the bow $E_{x1}$ at *v* = 10 kn and the longitudinal component of the noise at the stern $E_{x2}$ at *v* = 10 kn, and *v* = 15 kn.The results obtained from the tests allow us to state that:The high-speed motion of the platform caused stronger noise interference at the bow than that at the stern.Electric field noise signals were mainly composed of alternating components. The noise signals at the bow had the largest alternating components at *v* = 25 kn, whereas those at the stern had the smallest alternating components at *v* = 15 knThe effective value of noise at the bow was lower than that at the stern, implying that electric field sensors should be better placed near the bow.The electric field noise caused by the high-speed boat at different velocities was not white noise, so that it was not subject to a normal distribution and belonged to non-stationary random signals.


### 4.2. Frequency Domain Characteristics of Noise Signals

The target’s low frequency electric field was a line spectrum signal. For this reason, attention should be paid to both of its time and frequency domain characteristics in the noise analysis. Noise signals were time domain transformed to extract their frequency domain characteristics in this section. Calculations were conducted using the following procedure:Set the data length needed for Fast Fourier transform (FFT);Calculate the FFT of time sequence;Calculate the mean value of FFT amplitude spectrum squares;Draft the power spectrum curve;Calculate the cumulative energy distribution.

Following the above procedure, power spectra were calculated for the electric field noise signals collected from the motion platform at different velocities. The results are shown in Figure 6. The cumulative energy (area integral in energy spectrum estimation curve) was calculated to obtain the curve for the variation of cumulative energy and total energy with the frequency as shown in Figure 7.

In Figure 6 and Figure 7, it was found that:
The noise caused by the motion of the platform had its energy mostly at a frequency below 10 Hz, especially in the frequency bands below 2 Hz. The energy in the horizontal component of electric field noise at both the bow and stern exceeded 80% in the frequency bands below 2 Hz, and even reached 99%.The longitudinal component $E_{x1}$ and the transverse component $E_{y1}$ of the bow noise had more than 90% of the accumulated energy below 10 Hz, whereas the longitudinal component $E_{x2}$ and the transverse component $E_{y2}$ of the transom noise rose very slowly in the frequency band [10 Hz, 60 Hz] when the velocity reached 20 kn, and noticeably took up a lower proportion than the cumulative energy at other velocities. In other words, high-speed motion might cause large amounts of high frequency interference at the stern, which must be attributed to the joint effect of the engine’s electromagnetic radiation and the hull’s high-speed motion.


## 5. An Adaptive Low Frequency Electric Field Denoising Algorithm Based on ICEEMDAN with Threshold

As stated in Section 3, the electric field noise caused by high-speed motion belongs to non-stationary signals and its energy exists mainly at low frequencies. It overlaps the shaft frequency signals in certain frequency bands. Therefore, remote target detection depends on effective filtering of the noise signal. The use of an empirical mode decomposition algorithm (EMD) is one of the common methods often used to extract the characteristics of the non-stationary signal [29]. Using a complete ensemble empirical mode decomposition with adaptive noise, (ICEEMDAN) [30,31] can be an effective way to help overcome the modality mix problem in signal decomposition for the EMD algorithm and other improved EMD algorithms [32]. In this case, an adaptive denoising algorithm of the low frequency electric field based on ICEEMDAN with a threshold for the low frequency electric field noise is therefore proposed.

### 5.1. Procedure of Implementing the ICEEMDAN Algorithm

It was assumed that a noise signal was denoted by x(t) and decomposed in the ICEEMDAN to obtain the *k*th intrinsic mode function (IMF) $c_{k}(t)$ [33]. The residual was $r_{k}(t)$. The i(i=1,2,…,k) white Gaussian noise $s^{\left(i\right)}$ was added. The local mean value was M(⋅). Moreover, $E_{k}(\cdot )$ indicates the k(k=2,3,…,N) mode component obtained after the EMD. Thus, the flow chart for the procedure of implementing the ICEEMDAN algorithm is presented in Figure 8.

The procedure is detailed as follows:
Add the white Gaussian noise $s^{\left(i\right)}$ to the signal to be decomposed x(t) to obtain a new signal to be decomposed:(2)$$x{\left(t\right)}^{i}=x(t)+\beta_{o}E_{1}[s^{\left(i\right)}]$$Use an EMD algorithm to calculate the local mean value of $x{\left(t\right)}^{\left(i\right)}$, and take the mean value as the first residual $r_{1}(t)$:(3)$$r_{1}(t)=\frac{1}{K}\sum_{i=1}^{K}M[x{\left(t\right)}^{\left(i\right)}]$$Calculate the first mode IMF_1_ coefficient $c_{1}(t)$:(4)$$c_{1}(t)=x(t)-r_{1}(t)=x(t)-\frac{1}{K}\sum_{i=1}^{K}M[x{\left(t\right)}^{\left(i\right)}]$$Calculate the second mode IMF_2_ distribution $c_{2}(t)$ with Equations (3) and (4):(5)$$c_{2}(t)=r_{1}(t)-r_{2}(t)=r_{1}(t)-\frac{1}{K}\sum_{i=1}^{K}M\left\{r_{1}+\beta_{1}E_{2}[s^{\left(i\right)}]\right\}$$Calculate the *k*th IMF*_k_* distribution $c_{k}(t)$ in the similar way:(6)$$c_{k}(t)=r_{k-1}(t)-r_{k}(t)=r_{k-1}(t)-\frac{1}{K}\sum_{i=1}^{K}M\left\{r_{k-1}(t)+\beta_{k-1}E_{k}[s^{\left(i\right)}]\right\}$$Calculate the correlation coefficient of IMF distribution with the original signal $\alpha_{k}$. The greater the coefficient, the more real signals containing the target signal from the IMF decomposition. Meanwhile, all IMFs and the original signal were normalized to prevent removing the real IMFs with lower amplitude. The correlation coefficient of the *k*th IMF with the original signal x(t):(7)$$\alpha_{k}=\frac{\sum_{t=1}^{L}\left(x(t)-\bar{x})(c_{i}(t)-{\bar{c}}_{i}\right)}{\sqrt{\sum_{t=1}^{L}{\left(x(t)-\bar{x}\right)}^{2}\times \sum_{t=1}^{L}{\left(c_{i}(t)-{\bar{c}}_{i}\right)}^{2}\cdot }}$$
where L is the number of sampling points.


The correlation coefficient of the noise signals containing the target or not with IMF was calculated and varied with the decomposition layer as shown in Figure 9. 

Figure 9a,c,e,g shows the calculation results of autocorrelation coefficients of signals with targets by four sensors in the bow and stern of the ship. Figure 9b,d,f,h shows the results of autocorrelation coefficients calculated by four sensors, the bow and stern, on noise signals without targets, and provide a theoretical basis for threshold selection of subsequent algorithms through the comparison of autocorrelation coefficients. Because the analog source used in the experiment is 3 Hz, which is also based on the measured data, it is effective for 3 Hz. This paper designs a general method, because the frequency of the target radiation source is not limited to 3 Hz.

As revealed in Figure 9:
The target containing noise signals had fewer IMF decomposition layers than the noise signals containing no target, implying that the latter had a more complicated composition.The target containing noise signals had a larger correlation coefficient than the noise signals containing no target in some cases, so that denoising by filtering could be achieved through removing some correlation coefficients.The correlation coefficient of the target containing noise signals increased and then decreased with the increase of the decomposition layer. The correlation coefficient $\alpha_{4}$ was the largest at the fourth layer but became approximately 0 after reaching the seventh layer. This means that the signal at the fourth layer contained the largest amount of real information from the target source, so that it could be regarded as an approximately real signal. The signal after the seventh layer in the IMF decomposition contained very little information, so that it might be regarded as a ghost signal.The correlation coefficient of the noise signals containing no target increased with the decomposition layer. The correlation coefficient of the transverse component at the bow $E_{y1}$ with IMF contained two peaks and one trough when the velocity was less than 15 kn. It peaked at the second IMF layer and the sixth IMF layer but hit a trough at the fourth IMF layer. When the velocity exceeded 15 kn, the correlation coefficient varied in the same way as the other sensors; that is, increasing and then decreasing, peaking at the second IMF layer, and being regarded as a ghost signal after the ninth IMF layer.


### 5.2. An Adaptive Denoising Algorithm Based on the ICEEMDAN with Threshold

Threshold was calculated following such distribution of the correlation coefficients of the target containing noise signals and the noise signals containing no target as detailed in Section 4.1. The IMF screening was conducted while signals were reconstructed to denoise the target containing noise signals. It was assumed that the correlation coefficient of the kth IMF with the original signal was $\alpha_{k}$, and the threshold was TH. A set of targets containing noise signals x(t) was collected for the longitudinal component at the bow $E_{x1}$ when the velocity was 5 kn. The denoising procedure is as given in Figure 10.

The specific denoising procedure is as follows:
Perform the ICEEMDAN for the collected signals x(t) to output K IMF layers.Calculate the correlation coefficient of each IMF layer $\alpha_{k}(k=1,2,\ldots ,K)$;Set the threshold. Calculate the standard deviation of the correlation coefficient σ, and take it as the IMF screening threshold TH.Keep if $\alpha_{k}\geq \mathrm{TH}$;Screen the IMF. Keep the corresponding IMF*_k_* if $\alpha_{k}\geq \mathrm{TH}$, or discard it if not. Output $K_{1}$ is the screened IMF layers;Reconstruct the signal, superposed the $K_{1}$ IMF layers to obtain the denoised signal $x^{\prime }(t)$.


## 6. Verification of the Algorithm

Following the procedure of implementing the algorithm in Section 5, mode decomposition and adaptive IFM layer screening were conducted for the target containing noise signals collected by the sensors at different velocities. The results are given in Table 4.

Based on the IMF screening results in Table 4 and the denoising procedure, the collected signals as given in Section 2 were denoised. Line spectrum detection was also carried out for the reconstructed signals in the same way, to compare the results before and after filtering. The time sequence distribution diagram of the longitudinal component at the bow $E_{x1}$ at different velocities was drafted as shown in Figure 11. The left part of Figure 11 shows the calculated results of the signals containing the target, whereas the right part shows the calculation results of the signals containing no target. The comparison is presented in Table 5.

Based on Table 5, it is found that:
The proposed denoising algorithm could perform the adaptive layer screening and signal reconstruction based on the mode decomposition results of different original signals.The proposed algorithm achieved effective denoising. The line spectrum detection ratio and range were enhanced for all the signals collected by all the sensors at various velocities. The largest detection range exceeded 1300 m. Therefore, the algorithm was proven effective at denoising.By comparing the denoising algorithm in this paper with the standard ICEEDAN algorithm, the standard algorithm can also improve the line spectrum detection weight when the speed is low, but when the ship speed is higher than 10 kn, the denoising effect of the algorithm in this paper is significantly better than the standard ICEEDAN algorithm.A target could be always identified by the time domain distribution of signals after being filtered with the proposed algorithm regardless of the platform’s motion velocity.The proposed algorithm could overcome the interference caused by velocity, and ensure the remote target was detected during the high-speed motion of the platform. The line spectrum detection ratio of the longitudinal component at the bow $E_{x1}$ increased from 0% to 20.9%, and the detection range was expanded from 0 m to 821 m.The transverse component at the bow and stern had its detection ratio improved more than the longitudinal component at the bow and stern. This mirrored the larger transverse component on the route of the boat in motion in the previous analysis.


## 7. Conclusions

This paper illustrates the design of an electric field detection system with a high-speed boat as the platform, and the performing of an offshore test for its feasibility. Based on the line spectrum detection results, the sensors at the bow might experience a line spectrum detection failure when the velocity of the boat exceeded 20 kn. For this reason, a supplemental test was carried out to collect the interference noise generated by the boat in motion. Moreover, the noise characteristics of the platform at different velocities were analyzed in terms of time and frequency domains. The analysis revealed that the low frequency electric field noise affecting the detection performance of the high-speed motion platform belonged to non-stationary random signals, and the level of noise of the sensors at the bow was generally lower than those at the stern. This implied that sensors would be better if placed at the bow. The sensors at the bow and stern were equally effective when the velocity exceeded 15 kn. In addition, noise signals were mainly in low frequency bands.

To effectively filter the interference noise, an adaptive denoising algorithm based on ICEEMDAN with a threshold was proposed after considering the noise characteristics, and it was verified in simulation. The results proved that the proposed algorithm could effectively denoise the target containing noise signals collected at velocities of 5 kn, 10 kn, 15 kn, 20 kn and 25 kn. After being denoised, the signals had a distinct time domain distribution, and a higher signal to noise ratio. Moreover, the results of line spectrum detection were improved to different degrees for the denoised signals. The original line spectrum detection failed when the velocity exceeded 20 kn. After denoising, the detection range was improved by 20.9% from 0 m to 821 m, and the line spectrum detection of a target was successfully performed during high-speed motion. The total detection range of the proposed algorithm was increased by 53.1% from 853 m to 1306 m, to achieve remote target detection.

Future study will focus on improving the line spectrum detection method for further expansion of target detection range. Furthermore, a rigorous analysis in terms of capacity will be carried out on an unknown target.

## Figures and Tables

**Figure 1 sensors-22-05118-f001:**
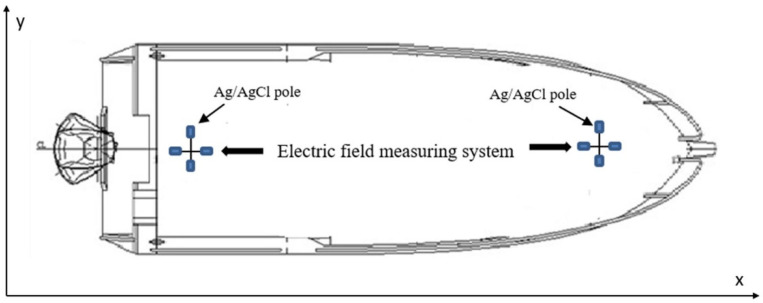
Schematic diagram of electric field measurement system based on fast moving platform.

**Figure 2 sensors-22-05118-f002:**
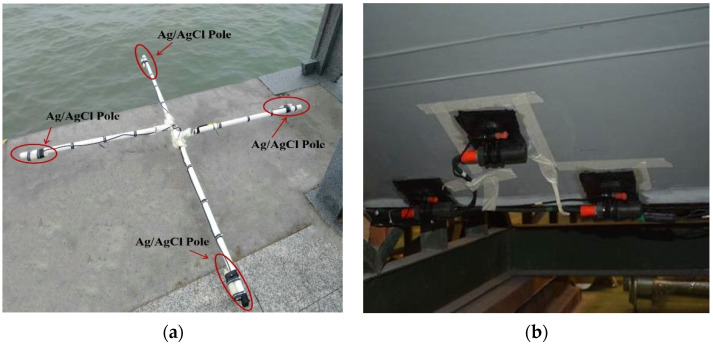
Actual construction drawing of measurement system.(**a**) a cruciform acquisition system based on a fixed bracket; (**b**) Actual sensor laying diagram.

**Figure 3 sensors-22-05118-f003:**
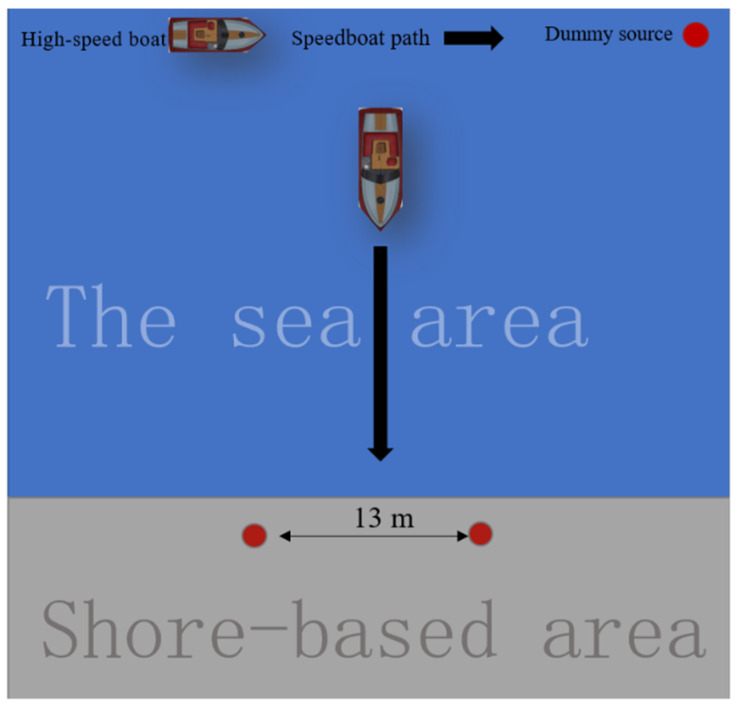
Schematic diagram of sea test: The speedboat moves from a distance to the simulated radiation source placed on the shore, the blue area is the sea area, the gray area is the shore the water depth in the test area was around 5 m.

**Figure 4 sensors-22-05118-f004:**
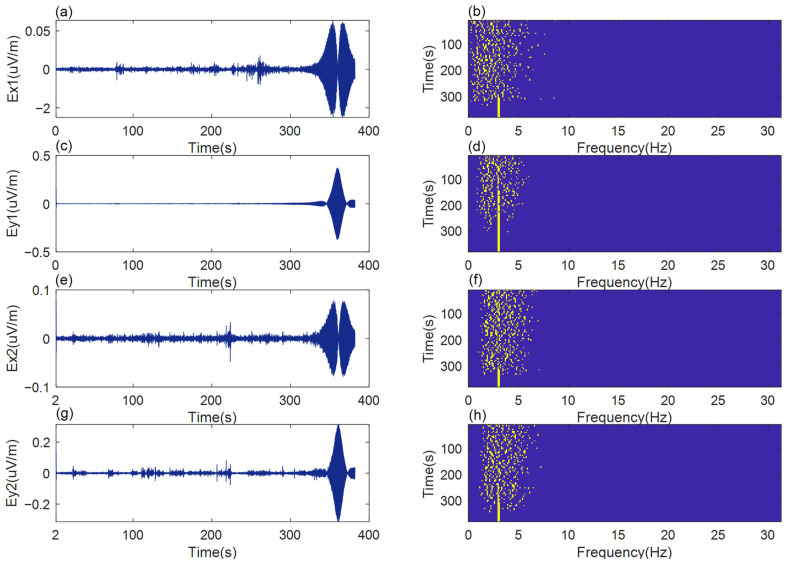
Low frequency electric field measurement and line spectrum detection at boat speed of 10 knots. (**a**) the longitudinal component of the bow sensor; (**b**) the result of target line spectrum extraction of *E_x_*_1_; (**c**) the transverse component of the bow sensor; (**d**) the result of target line spectrum extraction of *E_y_*_1_; (**e**) the longitudinal component of the stern sensor; (**f**) the result of target line spectrum extraction of *E_x_*_2_; (**g**) the transverse component of the stern sensor; (**h**) the result of target line spectrum extraction of *E_y_*_2_.

**Figure 5 sensors-22-05118-f005:**
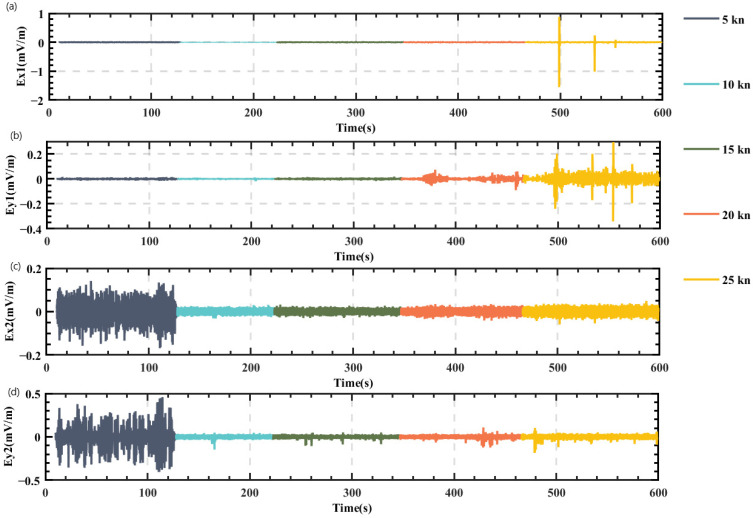
Time domain diagram of noise interference test caused by different platform velocity. (**a**) Longitudinal component of the bow sensor; (**b**) transverse component of the bow sensor; (**c**) longitudinal component of the stern sensor; (**d**) transverse component of the stern sensor.

**Figure 6 sensors-22-05118-f006:**
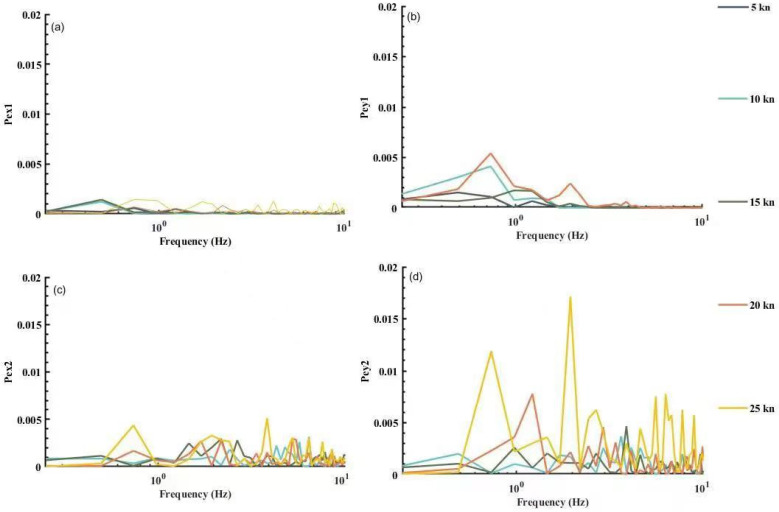
Noise power spectrum of the platform itself. (**a**) Power spectrum of the longitudinal component of the bow sensor; (**b**) Power spectrum of the transverse component of the bow sensor; (**c**) Power spectrum of the longitudinal component of the stern sensor; (**d**) Power spectrum of the transverse component of the stern sensor.

**Figure 7 sensors-22-05118-f007:**
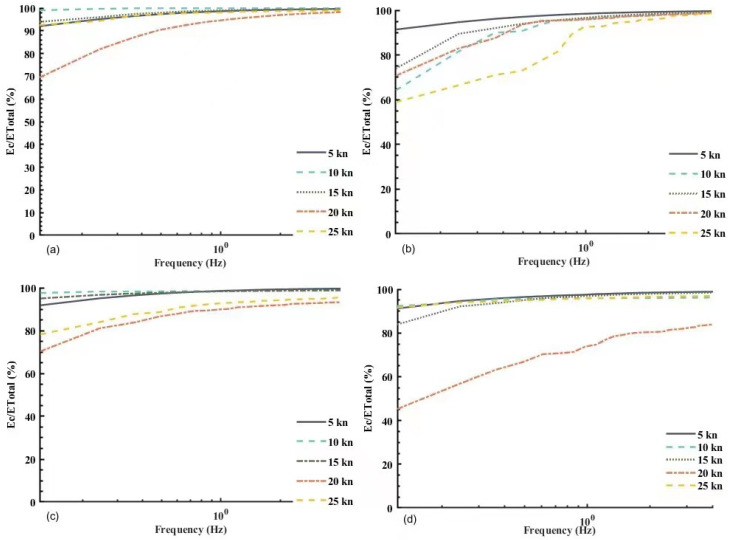
Distribution of accumulated energy with frequency. (**a**) Distribution of accumulated energy of the longitudinal component of the bow sensor; (**b**) Distribution of accumulated energy of the transverse component of the bow sensor; (**c**) Distribution of accumulated energy of the longitudinal component of the stern sensor; (**d**) Distribution of accumulated energy of the transverse component of the stern sensor.

**Figure 8 sensors-22-05118-f008:**
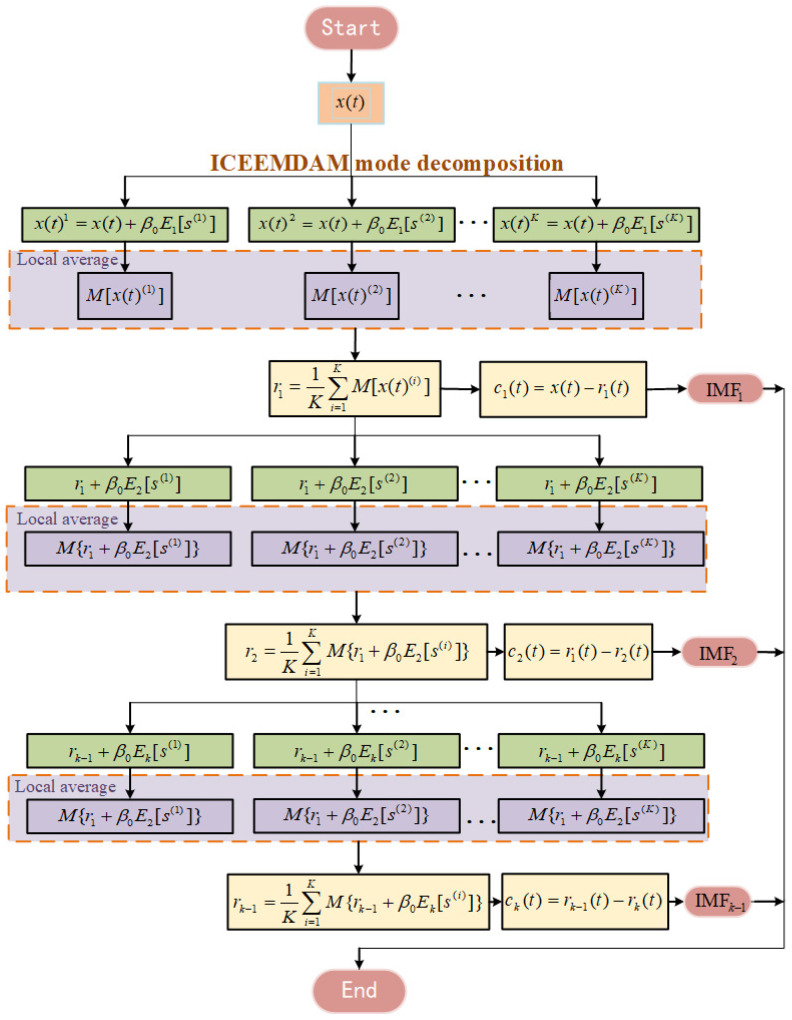
Flow chart of ICEEMDAN algorithm.

**Figure 9 sensors-22-05118-f009:**
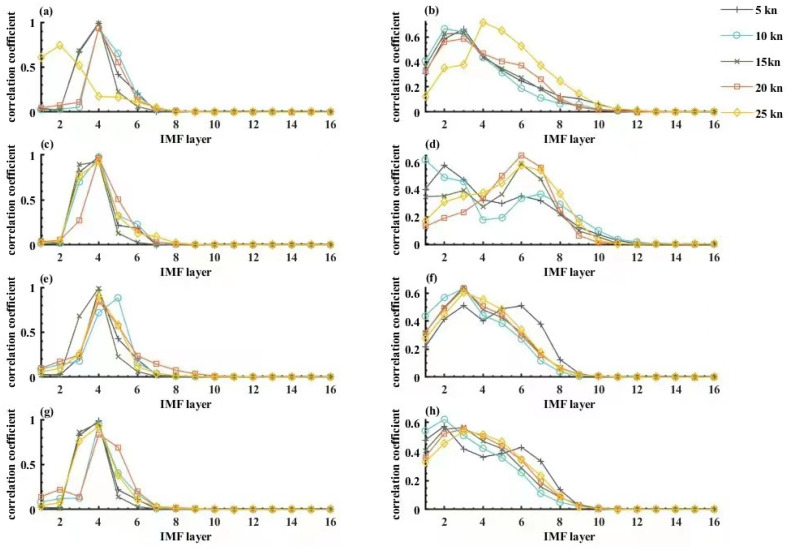
Correlation coefficients of IMF with and without targets as a function of decomposition layers.(**a**) Correlation coefficients for Ex1 at the bow; (**b**) Correlation coefficients for Ex1 at the stern; (**c**) Correlation coefficients for Ey1 at the bow; (**d**) Correlation coefficients for Ey1 at the stern; (**e**) Correlation coefficients for Ex2 at the bow; (**f**) Correlation coefficients for Ex2 at the stern; (**g**) Correlation coefficients for Ey2 at the bow; (**h**) Correlation coefficients for Ey2 at the stern.

**Figure 10 sensors-22-05118-f010:**
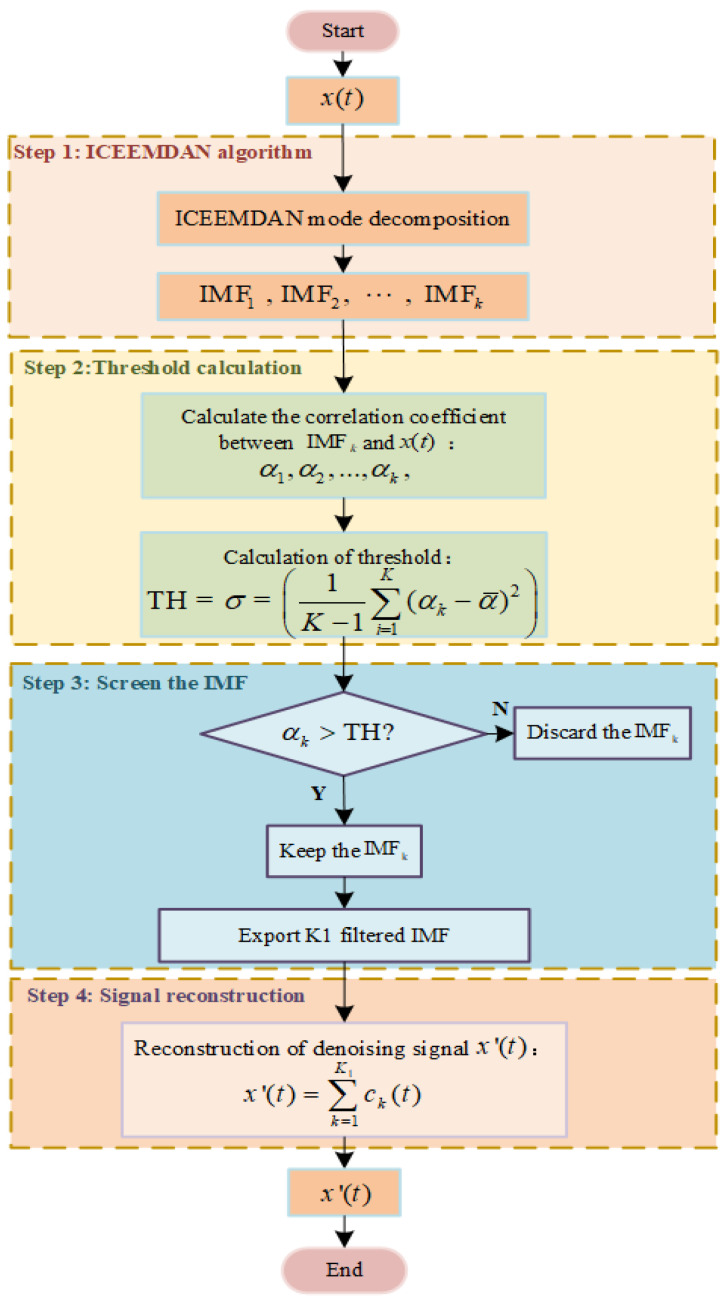
Flow chart of denoising procedure.

**Figure 11 sensors-22-05118-f011:**
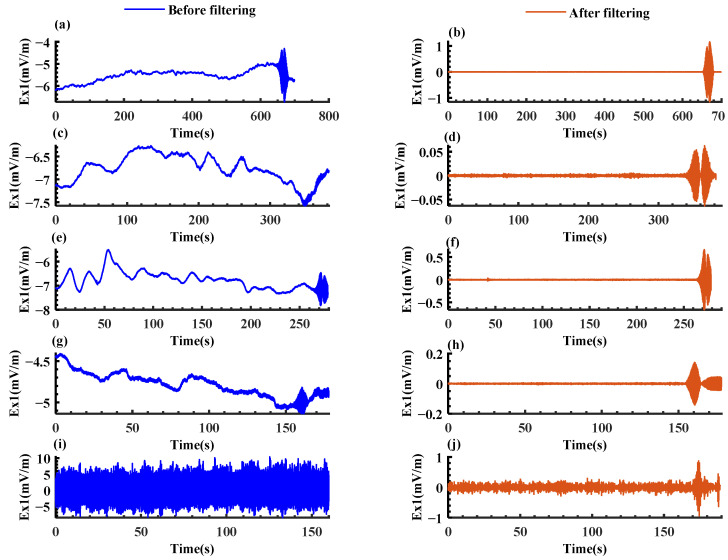
Temporal distribution comparison of bow longitudinal component $E_{x1}$ before and after filtering at different speeds. (**a**) Temporal distribution at 5 kn before filtering; (**b**) Temporal distribution at 5 kn after filtering; (**c**) Temporal distribution at 10 kn before filtering; (**d**) Temporal distribution at 10 kn after filtering; (**e**) Temporal distribution at 15 kn before filtering; (**f**) Temporal distribution at 15 kn after filtering; (**g**) Temporal distribution at 20 kn before filtering; (**h**) Temporal distribution at 20 kn after filtering; (**i**) Temporal distribution at 25 kn before filtering; (**j**) Temporal distribution at 25 kn after filtering.

**Table 1 sensors-22-05118-t001:** Parameter setting table of simulated radiation source.

Simulated Radiation Source	Electrode Spacing	Radiation Current	Radiation Frequency
MMO electrode	13 m	6.9 A	3 Hz

**Table 2 sensors-22-05118-t002:** Comparison of target electromagnetic field measurement results and line spectrum detection results when the measuring platform moves from low speed to high speed in turn.

$E_{x1}$					
*v* (kn)	5 kn	10 kn	15 kn	20 kn	25 kn
Testing frequency (Hz)	3 Hz	3 Hz	3 Hz	Failed to detect	Failed to detect
Testing time (s)	170 s	100 s	50 s	0 s	0 s
Testing distance (m)	439 m	514 m	386 m	0 m	0 m
Detect proportional *K*/%	44.7%	26.3%	13.2%	0%	0%
$E_{y1}$					
*v* (kn)	5 kn	10 kn	15 kn	20 kn	25 kn
Testing frequency (Hz)	3 Hz	3 Hz	3 Hz	3 Hz	3 Hz
Testing time (s)	330 s	130 s	60 s	60 s	45 s
Testing distance (m)	853 m	669 m	463 m	617 m	581 m
Detect proportional *K*/%	86.8%	34.2%	15.8%	15.8%	11.8%
$E_{x2}$					
*v* (kn)	5 kn	10 kn	15 kn	20 kn	25 kn
Testing frequency (Hz)	3 Hz	3 Hz	3 Hz	3 Hz	3 Hz
Testing time (s)	85 s	85 s	40 s	45 s	50 s
Testing distance (m)	219 m	437 m	308 m	465 m	646 m
Detect proportional *K*/%	22.4%	22.4%	10.5%	11.8%	13.2%
$E_{y2}$					
*v* (kn)	5 kn	10 kn	15 kn	20 kn	25 kn
Testing frequency (Hz)	3 Hz	3 Hz	3 Hz	3 Hz	3 Hz
Testing time (s)	110 s	100 s	45 s	50 s	65 s
Testing distance (m)	284 m	514 m	347 m	517 m	840 m
Detect proportional *K*/%	28.9%	26.3%	11.8%	13.2%	17.1%

**Table 3 sensors-22-05118-t003:** Calculation results of characteristic parameters.

$E_{x1}$					
*v* (kn)	5	10	15	20	25
$E_{p-p}$ (mV/m)	0.0181	0.0204	0.0391	0.0457	2.4327
μ (mV/m)	−3.3 × 10^−6^	7.3 × 10^−6^	−9.1 × 10^−7^	−7.7 × 10^−6^	5.8 × 10^−6^
$\sigma^{2}$ (mV^2^/m^2^)	1.0 × 10^−11^	1.8 × 10^−11^	2.2 × 10^−10^	6.4 × 10^−10^	4.1 × 10^−6^
$X_{rms}$ (mV/m)	0.0021	0.0024	0.0038	0.0051	0.0451
$K_{4}$	15.8 × 10^11^	8.6 × 10^10^	1.5 × 10^10^	4.6 × 10^9^	1.4 × 10^7^
$S_{f}$	1.2627	1.2548	1.2787	1.2701	5.2342
$E_{y1}$					
*v* (kn)	5	10	15	20	25
$E_{p-p}$ (mV/m)	0.0291	0.0338	0.0342	0.1644	0.6374
μ (mV/m)	−5.8 × 10^−6^	1.0 × 10^−5^	−5.1 × 10^−6^	3.6 × 10^−5^	−5.9 × 10^−5^
$\sigma^{2}$ (mV^2^/m^2^)	6.3 × 10^−7^	8.5 × 10^−12^	1.4 × 10^−10^	9.2 × 10^−9^	6.3 × 10^−7^
$X_{rms}$ (mV/m)	0.0033	0.0031	0.0036	0.0101	0.0283
$K_{4}$	2.8 × 10^10^	4.3 × 10^10^	1.8 × 10^10^	9.3 × 10^8^	2.4 × 10^6^
$S_{f}$	1.2787	1.2667	1.2654	1.5307	1.4974
$E_{x2}$					
*v* (kn)	5	10	15	20	25
$E_{p-p}$ (mV/m)	0.3102	0.2759	0.0617	0.0766	0.1087
μ (mV/m)	7.0 × 10^−5^	−6.3 × 10^−5^	3.3 × 10^−7^	3.6 × 10^−6^	−8.2 × 10^−6^
$\sigma^{2}$ (mV^2^/m^2^)	1.6 × 10^−6^	3.1 × 10^−8^	2.7 × 10^−9^	5.7 × 10^−9^	1.4 × 10^−8^
$X_{rms}$ (mV/m)	0.0356	0.0137	0.0072	0.0087	0.0111
$K_{4}$	2.1 × 10^6^	5.3 × 10^8^	1.1 × 10^9^	5.4 × 10^8^	2.4 × 10^8^
$S_{f}$	1.2876	1.2569	1.2517	1.2615	1.2514
$E_{y2}$					
*v* (kn)	5	10	15	20	25
$E_{p-p}$ (mV/m)	0.8632	0.7105	0.1569	0.2320	0.2884
μ (mV/m)	9.7 × 10^−5^	−9.2 × 10^−5^	1.1 × 10^−7^	−5.6 × 10^−7^	1.3 × 10^−5^
$\sigma^{2}$ (mV^2^/m^2^)	3.3 × 10^−5^	6.5 × 10^−7^	8.8 × 10^−9^	1.7 × 10^−8^	4.0 × 10^−8^
$X_{rms}$ (mV/m)	0.0761	0.0285	0.0097	0.0115	0.0142
$K_{4}$	1.8 × 10^5^	6.7 × 10^8^	6.8 × 10^8^	4.9 × 10^8^	2.6 × 10^8^
$S_{f}$	1.4408	1.3619	1.2808	1.3245	1.3215

**Table 4 sensors-22-05118-t004:** Screening results of IMF layers with target noisy signals collected by different sensors at different speeds.

$E_{x1}$					
v/(kn)	5	10	15	20	25
Layer number screening results	3–5	4–5	3–5	3–6	1–5
$E_{y1}$					
v/(kn)	5	10	15	20	25
Layer number screening results	3–5	3–6	3–4	3–5	3–5
$E_{x2}$					
v/(kn)	5	10	15	20	25
Layer number screening results	3–5	4–5	3–5	3–6	3–5
$E_{y2}$					
v/(kn)	5	10	15	20	25
Layer number screening results	3–5	4–5	3–4	2,4–5	3–5

**Table 5 sensors-22-05118-t005:** Comparison of line spectrum detection results before and after filtering.

$E_{x1}$						
v/(kn)		5	10	15	20	25
Specific gravity of line spectrum detection (%)	Before filtering	44.7	26.3	13.2	0	0
	After filtering by standard ICEEDAN	68.1	40.2	20.5	7.1	3.8
	After filtering of the algorithm in this paper	71.5	49.2	34.3	20.9	16.4
Detection distance of the spectrum (m)	Before filtering	439	514	386	0	0
	After filtering by standard ICEEDAN	668	785	599	278	186
	After filtering	702	966	1010	821	805
Distance improvement (%)	the standard ICEEDAN algorithm	+52.2	+52.7	+55.1	-	-
	Algorithm of this paper	+59.9	+87.9	+161.7	-	-
$E_{y1}$						
v(kn)		5	10	15	20	25
Specific gravity of line spectrum detection (%)	Before filtering	86.8	34.2	15.8	15.8	11.8
	After filtering by standard ICEEDAN	88.1	43.2	21.1	16.1	11.9
	After filtering of the algorithm in this paper	93.1	57.2	35.5	23.2	22.7
Detection distance of the spectrum (m)	Before filtering	853	669	463	617	581
	After filtering by standard ICEEDAN	865	845	618	628	585
	After filtering of the algorithm in this paper	914	1123	1046	911	1115
Distance improvement (%)	the standard ICEEDAN algorithm	+1.4	+26.3	+33.4	+1.7	+0.6
	Algorithm of this paper	+7.2	+67.9	+125.9	+47.6	+91.9
$E_{x2}$						
v(kn)		5	10	15	20	25
Specific gravity of line spectrum detection (%)	Before filtering	22.4	22.4	10.5	11.8	13.2
	After filtering by standard ICEEDAN	35.3	30.8	10.5	12.6	13.5
	After filtering of the algorithm in this paper	38.9	44.6	14.4	19.5	20.1
Detection distance of the spectrum (m)	Before filtering	219	437	308	465	646
	After filtering by standard ICEEDAN	347	600	308	496	660
	After filtering of the algorithm in this paper	382	876	424	766	987
Distance improvement (%)	the standard ICEEDAN algorithm	+58.4	+37.2	+0	+6.6	+2.1
	Algorithm of this paper	+74.4	+100.5	+37.7	+64.7	+52.8
$E_{y2}$						
v(kn)		5	10	15	20	25
Specific gravity of line spectrum detection (%)	Before filtering	28.9	26.3	11.8	13.2	17.1
	After filtering by standard ICEEDAN	49.3	33.9	20.1	15.5	17.8
	After filtering of the algorithm in this paper	53.3	41.0	38.9	28.9	26.6
Detection distance of the spectrum (m)	Before filtering	284	514	347	517	840
	After filtering by standard ICEEDAN	484	662	591	607	874
	After filtering of the algorithm in this paper	524	805	1131	1135	1306
Distance improvement (%)	the standard ICEEDAN algorithm	+70.4	+28.8	+70.3	+17.4	+4.0
	Algorithm of this paper	+84.5	+56.6	+225.9	+119.5	+55.5

## Data Availability

Not applicable.

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
