# Peer review of "Electric Field Detection System Based on Denoising Algorithm and High-Speed Motion Platform"

_sensors, 2022, doi:10.3390/s22145118_

Round 1
Reviewer 1 Report
1. The authors present an electric field measurement system based on a high-speed motion platform, which was constructed to analyze the characteristics of low frequency electric field noise. The proposed method can guarantee rapid and accurate target detection.
2. In the introduction, “[1],[2]” should be revised to [1-2]. The same mistake appears in other parts of the manuscript.
3. In the table 1, electrode spacing parameter, should be demonstrated in detail.
4. In the figure 5, Low frequency electric field measurement and line spectrum detection at boat speed of 10 knots, should be demonstrated in detail.
5. The manuscript has 13 figures; the number of the figures should be decreased.
6. Revise the English thoroughly before submission.
Author Response
Response to Reviewers 1
- The authors present an electric field measurement system based on a high-speed motion platform, which was constructed to analyze the characteristics of low frequency electric field noise. The proposed method can guarantee rapid and accurate target detection.
AUTHORS RESPONSE: The authors thank the reviewer for careful and timely suggestions. Every indication of the reviewer was taken into consideration by modifying the paper in the direction suggested, in this way the presentation of our work is certainly improved.
- In the introduction, “[1],[2]” should be revised to [1-2]. The same mistake appears in other parts of the manuscript.
AUTHORS RESPONSE: Thanks, the suggested revision has been performed for the entire document and has been marked yellow in the text.
- In the table 1, electrode spacing parameter, should be demonstrated in detail.
AUTHORS RESPONSE: Thanks for your valuable advice. Fig. 4 shows that the speedboat moves from a distance to the simulated radiation source set on the shore, the blue area is the sea area, the gray area is the shore, 13m indicates that the electric dipole moment of the simulated source is 13m, and the thick black arrow indicates that during the test, the speedboat moves from a distance to near the simulated source.
- In the figure 5, Low frequency electric field measurement and line spectrum detection at boat speed of 10 knots, should be demonstrated in detail.
AUTHORS RESPONSE: Thanks for your valuable advice. It is mentioned in line 148 of the paper that the line spectrum extraction method used is the method adopted in literature [28]. The line spectrum extraction process is only used as the last means to verify the denoising algorithm proposed in this paper. Readers can obtain the line spectrum detection method through literature [28], so this paper does not give a detailed description. A measurement system carried by the boat as the platform could achieve the effective detection of a target’s low frequency electric field signals. The target’s line spectrum could be successfully extracted from the electric field signals collected at the bow and stern, verifying the feasibility of the low frequency electric field detection with a high-speed boat as the platform.
- The manuscript has 13 figures; the number of the figures should be decreased.
AUTHORS RESPONSE: Thanks for your valuable advice. After careful consideration and repeated discussion, all the authors of this paper believe that the number of pictures in this paper has been controlled to the minimum. All figures have been revised and improved where necessary. It was not possible to reduce the number of figures because the paper was structured on the basis of the information presented in the figure.
- Revise the English thoroughly before submission.
AUTHORS RESPONSE: Thanks for your valuable advice. A complete revision of the English has been carried out.

Reviewer 2 Report
The Authors present an electric field measurement system based on a high-speed boat along with a denoising algorithm to reduce the intrinsic noise generated by the system itself. The manuscript is well written although some sentences should be corrected. Overall the manuscript presents a scientifically sound experiment and an algorithm which can be of interest to the reader.
Minor issues:
-please correct the sentences at lines 65-66 and caption Fig3
-labels of the figures should be made more readable.
Please see attached file for more comments

Author Response
Response to Reviewers 2
The Authors present an electric field measurement system based on a high-speed boat along with a denoising algorithm to reduce the intrinsic noise generated by the system itself. The manuscript is well written although some sentences should be corrected. Overall the manuscript presents a scientifically sound experiment and an algorithm which can be of interest to the reader. The manuscript is in my opinion suitable for publication provided that some minor issues are addressed:
AUTHORS RESPONSE: The authors thank the reviewer for careful and timely suggestions. Every indication of the reviewer was taken into consideration by modifying the paper in the direction suggested, in this way the presentation of our work is certainly improved. reader.
Minor issues:
- please correct the sentences at lines 65-66 and caption Fig3
AUTHORS RESPONSE: Thanks for the expert's valuable advice. Corrections have been made to the sentence and caption.
The specific modification is as follows:
- labels of the figures should be made more readable.
AUTHORS RESPONSE: Thanks for the expert's valuable advice. A thorough review of the image labeling was carried out.
- line 46: please explain clearly the terms XRay and ZRay.
AUTHORS RESPONSE: Thanks for the expert's valuable advice. The whole sentence has been revised to make it more understandable.
The specific modification is as follows:
Moore et al. [13], performed a test with Seaglider, an underwater acoustic glider de-veloped in the United States, to collect the calls of blue whales, humpback whales and sperm whales. In this US project, omnidirectional broadband hydrophones (5 Hz to 30 kHz) were mounted on the wings of the glider.
- line 128: the caption is not clear. Please explain.
AUTHORS RESPONSE: Thanks for the expert's valuable advice. The caption has been revised.
The specific modification is as follows:
A low frequency electric field detection test with a high-speed boat as the platform
- Figure5: the labels are too small and difficult to read.
AUTHORS RESPONSE: Thanks for the expert's valuable advice. Figure 5 has been modified according to reviewer suggestions
- Figure 7: the labels are too small and difficult to read.
AUTHORS RESPONSE: Thanks for the expert's valuable advice. Figure 7 has been modified according to reviewer suggestions
- Figures 8 and 9: the labels of the axes and the legends are too small and difficult to read
AUTHORS RESPONSE: Thanks for the expert's valuable advice. Figure 8,9 have been modified according to reviewer suggestions.
- Equation 2 to 6 lines 300- 307. Coefficients should be aligned with the text
AUTHORS RESPONSE: Thanks for the expert's valuable advice. Equation 2 has been revised.
- Figure 11: axes labels are too small and difficult to read
AUTHORS RESPONSE: Thanks for the expert's valuable advice. Figure 11 has been modified according to reviewer suggestions
- Performance of the proposed algorithm could be compared to standard algorithm for denoising in order to assess its capabilities.
AUTHORS RESPONSE: A comparison of line spectrum detection results before and after filtering has been presented in table 5. By adding a comparison of the denoising effect with the standard ICEEDAN algorithm and adding the filtering effect of the standard ICEEDAN algorithm in Table 5, respectively, and the specific percentage of distance improvement.

Reviewer 3 Report
The manuscript "Characteristics of Low Frequency Electric Field Noise from a High-Speed Boat and a Denoising Algorithm " applied ICEEMDAN combined with an adaptive threshold algorithm to denoising low-frequency noises from a high-speed boat. The results seem that the algorithm well improved the SNR. However, there are so many errors in text, figures, and equations should be modified to make the manuscript more readable. Moreover, the demonstration of the advantage of this algorithm over other methods is too weak. Detailed comments are listed as following.
1) There is two “correspondence”. The email seems not belong to the first author, rather than the corresponding author.
2) The title should be revised to emphasize the highlight of this manuscript.
3) ICEEMDAN is not a new method, current research status should be stated to emphasize the research gap.
4) The red text or numbers should be avoided.
5) The format of all the characters should be the same in the whole manuscript.
6) When different results are compared, such as Figure 8, the same coordinate should be used. In addition, (a~d) should be referred specifically.
Author Response
Response to Reviewers 3
The manuscript "Characteristics of Low Frequency Electric Field Noise from a High-Speed Boat and a Denoising Algorithm " applied ICEEMDAN combined with an adaptive threshold algorithm to denoising low-frequency noises from a high-speed boat. The results seem that the algorithm well improved the SNR. However, there are so many errors in text, figures, and equations should be modified to make the manuscript more readable. Moreover, the demonstration of the advantage of this algorithm over other methods is too weak. Detailed comments are listed as following.
- There is two “correspondence”. The email seems not belong to the first author, rather than the corresponding author.
AUTHORS RESPONSE: Thank for the careful review and valuable comments. The authors' emails have been revised
- The title should be revised to emphasize the highlight of this manuscript.
AUTHORS RESPONSE: Thank for the careful review and valuable comments. The title has been revised as suggested by reviewer.
The specific modification is: Electric field detection system based on Denoising Algorithm and High-Speed motion platform
- ICEEMDAN is not a new method, current research status should be stated to emphasize the research gap.
AUTHORS RESPONSE: Thank for the careful review and valuable comments. The reviewer is right, ICEEMDAN is not a new method, in fact the authors already declare from the abstract that this algorithm has been improved and combined with an adaptive threshold algorithm for denoising and reconstructing target containing noise signals.
- The red text or numbers should be avoided.
AUTHORS RESPONSE: Thank for the careful review and valuable comments. Red text and numbers have been removed
- The format of all the characters should be the same in the whole manuscript.
AUTHORS RESPONSE: The format of the entire document was revised
- When different results are compared, such as Figure 8, the same coordinate should be used. In addition, (a~d) should be referred specifically.
AUTHORS RESPONSE: Figure 8 has been modified according to reviewer suggestions

Round 2
Reviewer 1 Report
no further comment.
Author Response
Thanks for the expert review of the article, and hats off to you for your dedication!
Reviewer 3 Report
Although the authors claimed that the manuscript has been revised according to the the previous comments, some simple but important questions have not been not answered.
First of all, there are two “correspondence”. The email seems belong to the first author, rather than the corresponding author. This could be a serious ethic problem, because all the authors should read the whole manuscript and confirm their information. As the corresponding email is belong to the first author, I quite doubt whether the corresponding author know the submitted manuscript.
Secondly, there are still many many errors in text, figures, and equations, which should be modified to make the manuscript more readable. For instance, "Grund et ICEEMDAN al. [14] used" in Line 47; format inconsistency in Line 137; figures with very poor quality.
Author Response
Response to Reviewer 3
1、First of all, there are two “correspondence”. The email seems belong to the first author, rather than the corresponding author. This could be a serious ethic problem, because all the authors should read the whole manuscript and confirm their information. As the corresponding email is belong to the first author, I quite doubt whether the corresponding author know the submitted manuscript.
AUTHORS RESPONSE:
Thank for the careful review and valuable comments.
We are sorry for such a mistake, and based on your suggestion, the email address of the corresponding author has been reworked and revised to jiang_runxiang@163.com, in line with the submission system.
In future submissions, we will pay close attention to this detail and not make such mistakes again.
Once again, thank you and salute your careful review and hard work!
2、Secondly, there are still many many errors in text, figures, and equations, which should be modified to make the manuscript more readable. For instance, "Grund et ICEEMDAN al. [14] used" in Line 47; format inconsistency in Line 137; figures with very poor quality.
AUTHORS RESPONSE:
Thank for the careful review and valuable comments.
Based on your suggestions, we have carefully reorganized and revised the entire text, and have touched up the language expressions in a comprehensive manner.
In order to make it easier for experts to read, the revised parts are highlighted in yellow, and the errors of expression in the article have been corrected one by one, and each picture has been revised to improve the quality of the pictures to an acceptable level.
Once again, thank you for your careful review. Your valuable suggestions are our motivation for continuous improvement, hats off to you!
Round 3
Reviewer 3 Report
Thank you for your effort to provide the revised manuscript again. I appreciate your hard work and patience to improve your paper. It is a quiet meaningful work you did. Still, I think there are several issues should be addressed before publication.
1) Overall, there are some errors in format or text. For instance, several words are capitalized; Error! Reference source not found in Line 54; lack of description of (a-d) in Figure 7;
2) x and y in Ex and Ey should be labeled in Figure 1 to make it easy to understand.
3) Please give some explanation why the noise of Ey1 is smaller than others in Figure 4.
4) Line 174, it is difficult to the meaning of “The maximum detection time for a target’s line spectrum was Tmax”. The definition of Tmax should be rephrased and labelled in Figure 4.
5) Is the data at the acceleration stages (such as from 5kn to 10kn) also considered in Figure 5. Please clarify whether the acceleration will make any difference.
6) Line 270, “the energy of the horizontal component of the noise at the stern Ex2 and Ey2” should be “…Ex1 and Ex2”? Moreover, in this part, only Ex was discussed and no discussion is made on Ey.
7) Figure11, please clarify why the signals (before filtering) are different from Figure 4.
8) Is duplicated test conducted in this study? It is important when you concluded the distance improvement in Table 5. As the improvements vary for different components and different velocities, I think it not appropriate to calculate increase of the total detection range.
9) Is there any influence of the source frequency on the performance of the proposed method? Only 3Hz was discussed and it is very close to the frequency of noise, as stated by the authors that “The noise caused by the motion of the platform had its energy mostly at a frequency below 10Hz, especially in the frequency bands below 2Hz”.
Author Response
Response to Reviewer 3 Thank you for your effort to provide the revised manuscript again. I appreciate your hard work and patience to improve your paper. It is a quiet meaningful work you did. Still, I think there are several issues should be addressed before publication. AUTHORS RESPONSE: The authors thank the reviewer for careful and timely suggestions. Every indication of the reviewer was taken into consideration by modifying the paper in the direction suggested, in this way the presentation of our work is certainly improved. 1、Overall, there are some errors in format or text. For instance, several words are capitalized; Error! Reference source not found in Line 54; lack of description of (a-d) in Figure 7; AUTHORS RESPONSE: Thank for the careful review and valuable comments and I am sorry for my carelessness. The reference links have been fixed and the Figure 7 caption description has been improved. 2、x and y in Ex and Ey should be labeled in Figure 1 to make it easy to understand. AUTHORS RESPONSE: Thank for the careful review and valuable comments. Based on your suggestions, x and y in Ex and Ey have been labeled in Figure 1. 3、Please give some explanation why the noise of Ey1 is smaller than others in Figure 4. AUTHORS RESPONSE: Thank for the careful review and valuable comments. The following explanation was added to the paper: At a speed of 10 knot, the speed of the boat is not high, and the speed of the water flow is mainly along the longitudinal direction: The transverse electric field in the bow is away from the engine, which is placed in the stern, so there is no interference caused by the splashing of the water due to the high speed of the boat. Therefore, the noise is minimal, and the detection result is the best. 4、 Line 174, it is difficult to the meaning of “The maximum detection time for a target’s line spectrum was Tmax”. The definition of Tmax should be rephrased and labelled in Figure 4. AUTHORS RESPONSE: Thank for the careful review and valuable comments. The following explanation was added to the paper: With Tmax we specify the maximum time necessary to effectively detect the line spectrum of the target. Since the spectrum of the lines is not always continuous, it can be divided into several fragments. The sum of these contributions of the detection times gives the maximum total time necessary to correctly detect the target. 5、 Is the data at the acceleration stages (such as from 5kn to 10kn) also considered in Figure 5. Please clarify whether the acceleration will make any difference. AUTHORS RESPONSE: Thank for the careful review and valuable comments. In response to your question, I thought carefully and answered as follows: This article mainly studies the characteristics of the electric field and the denoising algorithm at different boat speeds. Therefore, in order to conveniently observe the time domain characteristics, Figure 5 integrates the data collected at different ship speeds in the same time axis. Since the actual acceleration process is not measured uniformly and the acceleration process is different, the acceleration variation process is not considered and studied in this paper. Also, the target moves at a constant speed under normal circumstances, so this article focuses on studying smooth motion at different speeds. In the future, the influence of the acceleration variation can be further studied. 6、 Line 270, “the energy of the horizontal component of the noise at the stern Ex2 and Ey2” should be “…Ex1 and Ex2”? Moreover, in this part, only Ex was discussed and no discussion is made on Ey. AUTHORS RESPONSE: Thank for the careful review and valuable comments. In response to your question, I thought carefully and answered as follows: Firstly, we explain what we want to express there: here we mainly want to express that the electric field noise of high-speed speedboats will have more high-frequency components at the stern and less low-frequency interference, but there are parts that are incomplete when the text is described and can easily lead to misunderstanding by readers, so we have reworked it according to the experts' revision. Modified parts are marked in yellow in the text 7、Figure11, please clarify why the signals (before filtering) are different from Figure 4. AUTHORS RESPONSE: Thank for the careful review and valuable comments. In response to your question, I thought carefully and answered as follows: Based on the target acquisition, this document separately collects the noise at different boat speeds, studies its characteristics and proposes a filtering algorithm based on the noise characteristics. Figure 11 shows the signal acquisition with the target source at different boat speeds and Figure 4 shows the pure noise acquisition with no target source at different boat speeds, so the two signals are different. An explanation was added to the paper. 8、Is duplicated test conducted in this study? It is important when you concluded the distance improvement in Table 5. As the improvements vary for different components and different velocities, I think it not appropriate to calculate increase of the total detection range. AUTHORS RESPONSE: Thank for the careful review and valuable comments. In response to your question, I thought carefully and answered as follows: In this paper, for different speeds of the target detection in the test to ensure that the same time of the target movement at different speeds, so that the actual movement of different speeds of different distances, the total distance of movement of fast speed is far. However, when comparing the results, the results are compared between the same motion speed, and the line spectrum detection weight as well as the relative detection distance is increased for the same speed before and after the signal processing, therefore, after careful discussion by all the authors, the before and after comparison performed in the analysis of the results is feasible. At the same time, in table 5, to compare the effectiveness of this algorithm, the ICEEMDAN algorithm is used to decompose and reconstruct the signal. The test distance (m) in the table expresses the detection distance of the spectrum, I changed the label to avoid misunderstandings. What I want to express here is that the spectrum of the target lines can effectively detect the distance after signal processing in three different situations, i.e. before filtering, after applying the general ICEEMDAN algorithm and after applying the algorithm presented in this article. 9、Is there any influence of the source frequency on the performance of the proposed method? Only 3Hz was discussed and it is very close to the frequency of noise, as stated by the authors that “The noise caused by the motion of the platform had its energy mostly at a frequency below 10Hz, especially in the frequency bands below 2Hz”. AUTHORS RESPONSE: Thank for the careful review and valuable comments. After careful discussion and reflection, we agreed that the frequency of the source has little effect on the performance of the algorithm. The 3Hz electric field signal used in this paper is mainly the fundamental frequency of the boat target's axial frequency electric field. The other frequency bands of the axial frequency electric field are multiples of the fundamental frequency of 3Hz, this is because it is very close to the frequency band below 2Hz. The actual extraction of the signal into the frequency band will provide a reference for the subsequent extraction of the remaining higher frequency bands. If the problem can be solved and the 3Hz frequency band can be extracted, signals above the 3Hz frequency band can also be extracted more effectively. This method aims to simplify the way to the study of the target electric field of boats. Since there are some difficulties in conducting marine experiments, this article only conducts data acquisition and related research on the simulated 3Hz radiation source. In the next step, experiments with different frequencies may also be carried out to verify the efficacy of the method. Once again, thank you for your careful review. Your valuable suggestions are our motivation for continuous improvement, hats off to you!
